# Electrical Sensor Calibration by Fuzzy Clustering with Mandatory Constraint

**DOI:** 10.3390/s24103068

**Published:** 2024-05-11

**Authors:** Shihong Yue, Keyi Fu, Liping Liu, Yuwei Zhao

**Affiliations:** School of Electrical Engineering and Automation, Tianjin University, Tianjin 300072, China; shyue1999@tju.edu.cn (S.Y.); fukeyi@tju.edu.cn (K.F.); zyww@tju.edu.cn (Y.Z.)

**Keywords:** sensor, electrical tomography, calibration, mandatory constraint, fuzzy clustering

## Abstract

Electrical tomography sensors have been widely used for pipeline parameter detection and estimation. Before they can be used in formal applications, the sensors must be calibrated using enough labeled data. However, due to the high complexity of actual measuring environments, the calibrated sensors are inaccurate since the labeling data may be uncertain, inconsistent, incomplete, or even invalid. Alternatively, it is always possible to obtain partial data with accurate labels, which can form mandatory constraints to correct errors in other labeling data. In this paper, a semi-supervised fuzzy clustering algorithm is proposed, and the fuzzy membership degree in the algorithm leads to a set of mandatory constraints to correct these inaccurate labels. Experiments in a dredger validate the proposed algorithm in terms of its accuracy and stability. This new fuzzy clustering algorithm can generally decrease the error of labeling data in any sensor calibration process.

## 1. Introduction

Various sensors play an important role in detection processes in industry, and almost all sensors must be calibrated before they can be used in formal applications [1]. Different sensors have different calibration methods. The characterization and low-cost calibration of particulate matter sensors were proposed at a high temporal resolution to a reference-grade performance, and the frequencies and duration were tested at a 2 min resolution [2]. A novel multilocation calibration scheme was introduced specifically to target mobile devices, and the scheme exploited machine learning techniques to perform an adaptive, power-efficient auto-calibration procedure through which it achieved a high level of output sensor accuracy when compared to that of state-of-the-art techniques [3]. An on-site sensor calibration method was proposed for the quality assurance of process separation measurements, which can guarantee the optimal performance of the sensor measuring system and assure a high measurement quality between company inspections [4]. More reviews can be found in [5,6,7].

Due to its advantages of being nonradiative, non-invasive, and low cost, as well as having fast responses, electrical tomography (ET) [8] has been widely used in industrial detection processes. Accordingly, ET sensors (ETSs) [9,10] are ever-increasingly used for parameter detection for multiphase flow in pipes, such as the solid-phase fraction (SPF), flow velocity, and flow regime, etc. In this study, we focus on the measurements and calibrations of ETSs when detecting the SPF for two-phase solid–liquid flow [11]. In our previous study [12], a calibration method was proposed when an ETS was used to detect the flowing velocity. However, when an ETS is used to detect different SPFs, its calibration is very difficult due to various flow patterns and complex measuring conditions.

ETS calibration can be categorized into three types: ex-factory calibration, indirect calibration from other sensors, and direct calibration from sampling data. Indirect calibration can be performed within various measuring conditions and represent all the working conditions that ETSs operate in. But these calibrating data may be erroneous and inaccurate. Inversely, both ex-factory and sampling data are accurate, but they cannot fully reproduce and represent all actual measuring conditions. According to the case-based reasoning (CBR) principle [13], “similar problems must have similar solutions”. And if any two measurements are similar, their labels must be consistent, and inversely, two different measurements should have different labels. Hence, a set of similar measurements must be distributed in a cluster within which any two points are close together, and unsimilar measurements must belong to different clusters. Any clustering algorithm can find various data distributions or clusters [14]. Accordingly, similar measurements from ETSs have the same cluster label whereas dissimilar ones have different labels. Consequently, the actual measurements from indirect data in ETSs have a clustering structure [15], and any clustering algorithm can find the data distribution. It is always possible to obtain a portion of special data with accurate labels, which can form mandatory constraints to correct labeling errors in other data.

Due to the inconsistent and uncertain characteristics of inaccurate labeling data, they can be represented as the fuzziness in a fuzzy clustering algorithm [16], such as the most common one, fuzzy c-means (FCM) clustering [17]. In this paper, we propose a semi-supervised fuzzy clustering algorithm that takes the fuzzy membership degree of these special data as a set of mandatory constraints, reestablishes the objective function, and performs alternating optimization to achieve a clustering analysis of all the historical data used for the calibration. By using the fuzzy membership degree with and without mandatory constraints as variables, all data labels are reclassified and calibrated. When using the SPF as the label, the calibrated new label is introduced into the most commonly used SPF algorithm, the linear regression algorithm [18], to compare the accuracies of the two labels before and after the calibration.

## 2. Related Work

This section includes the ETS principle, the SPF calculation, and the FCM algorithm.

### 2.1. ETS and SPF Calculation

We use a typical 16-electrode ET system to explain the ETS’s measuring principle. The ETS measures the SPF in a field Ω by boundary measurements [19]. Figure 1a shows the ETS measuring process in Ω. First, an exciting current “I” is added to electrode 1, and 15 measurements are obtained in 15 other electrodes. Then “I” is added to electrode 2, and 15 measurements are obtained again. The process is repeated in turn until all 16 electrodes are excited. Therefore, a total of 240 obtained measurements are used to construct 16 U-shaped curves, in which each responds to the same excitation, as shown in Figure 1b.

On the basis of prior information and for the repeatability of various SPFs during the working process, to perform the SPF calculation, we take the vector with 240 measurements as an input variable, and the corresponding label of the SPF as the output variable. The relation *f*(·) from the input to the output is characterized as follows:(1)$$f:X\to \eta =f(X),\;s.t.,\;X\in R^{240},\eta \in R^{1}$$

A set of prior historical data pairs (input *X*, output *η*) in $(X_{k},\eta_{k}{)}_{\left(0\leq k\leq n\right)}$ are fitted with either global or piecewise linear formulas for the SPF. Denoting *E* as the unit vector, the relationship from *X* to *η* is assumed to be approximately linear, so that it can be expressed by the parameters *a* and *b* as follows:(2)η=f(X)=aX+bE

Generally, there are no parameters *a* and *b* that exactly satisfy the equation by $(X_{k},\eta_{k}{)}_{\left(0\leq k\leq n\right)}$. Let *X*′ = [*X E*], *C* = (*a b*)*^T^*. A common approach is to use the least squares method to solve the following optimization problem:(3)$$\min z=\sum_{k=1}^{n}||\eta_{k}-(aX_{k}+bE)|{|}_{2}$$

Based on the Joseph-Louis Lagrange’s criterion [20], Equation (3) has an analytic solution as follows:(4)$$c={\left(X^{\prime T}X^{\prime }\right)}^{-1}X^{\prime T}\eta$$

However, to reduce the over-fitting effect and noise, it is usually necessary to add a regularization parameter *λ* to obtain the following regularization solution:(5)$$c={\left(X^{\prime T}X^{\prime }+\lambda E\right)}^{-1}X^{\prime T}\eta$$

When the relation *f*(·) is highly nonlinear, piecewise linear fitting is required as shown below:(6)$$c_{s}={\left(X_{s}^{\prime T}X_{d}^{\prime }+\lambda E\right)}^{-1}X_{s}^{\prime T}\eta_{s}$$
where $\eta_{s}\in [I_{s},I_{s+1}]$, *s* = 1, 2, …, *M*, and [*I_s_*, *I_s_*_+1_] is divided into *M* intervals according to *η_s_*; however, due to the complexity of working conditions, it is necessary to analyze the applicable range of the above calculation method.

### 2.2. FCM Clustering Algorithm

Let *S* = {*x_i_*|*i* = 1, 2, …, *n*} be a dataset with *n* data vectors distributed in *c* clusters, *x_i_*∈*R^d^* in a *d*-dimensional data space. The typical fuzzy clustering algorithm’s FCM is reviewed as follows. The objective function in the FCM can be stated as follows:(7)$$\min J(U,V)=\sum_{i=1}^{c}\sum_{j=1}^{n}u_{ij}^{m}d_{ij}^{2},\;s.t.\;\sum_{i=1}^{c}u_{ij}=1,\;j=1,2,\ldots ,n,0<\sum_{j=1}^{n}u_{ij}\leq n,$$
where $d_{ij}=||x_{j}-v_{i}||$, *v_i_* is the prototype (center) of the *i*th cluster, *u_ij_* is the membership degree of the *j*th vector to the *i*th cluster, and *m* is a fuzziness exponent, ranging in the interval of [1,3].

Using Lagrange multiplier optimization [21], both *u_ij_* and *v_i_* in Equation (7) can be solved as follows:(8)$$u_{ij}={\left(\sum_{r=1}^{c}d_{ij}^{2/(m-1)}/d_{rj}^{2/(m-1)}\right)}^{-1}\;\mathrm{and}\;v_{i}=\sum_{j=1}^{n}u^{m}{\;}_{ij}x_{j}/\sum_{j=1}^{n}u^{m}{\;}_{ij}$$

All fuzzy membership degrees consist of an *n* × *c* partition matrix *U* = [*u_ij_*]. The steps of the FCM are shown in Algorithm 1. But the FCM cannot utilize any a priori information in practice [22,23]. This information is not only helpful for boosting the clustering quality but also for meeting mandatory application requirements. In this paper, we proposed a new method to address these problems along a solid mathematical optimization process.
**Algorithm 1. The FCM algorithm**.Input: Dataset *S*, the number of clusters *c*, exponent indexes *m*, and acceptable error *ε*Output: The clustering label of each datum in *S*Method:(1) Initialize all clustering centers in FCM as *v*_1_, *v*_2_, …, *v_c_*;(2) Problem 1:   Fix *v_i_* and solve *u_ij_* by the first formula in Equation (8), *i* = 1~*c*, *j* = 1~*n*; (3) Problem 2:   Fix *u_ij_* and solve *v_i_* using the second formula in Equation (8), *i* = 1~*c*;(4) Stop if the difference of the partition matrix at the *t*th iteration satisfies ||*U^t^*^+1^-*U^t^*|| ≤ *ε* and go to Step (5); otherwise, go to Step (2);(5) Partition *S* into c clusters: *C*_1_, *C*_2_, …, *C_c_* by the fuzzy membership degrees of all data. 

## 3. Mandatory Constraint-Based Fuzzy Clustering for Decreasing Error in Inaccurate Data

In this section, a new fuzzy clustering algorithm is proposed to decrease the error in inaccurate calibration data after introducing these typical data types from an ETS in practice.

### 3.1. Three Types of Calibration Data

The three types of calibration data for an ETS are explained separately.

(1) *Ex-factory calibration data*. The ex-factory calibration process of an ETS is shown in Figure 2. The ETS is connected to a data acquisition device, and a group of rods with the same diameter and length are vertically inserted into the cross-sectional ETS. Each group of rods responds to a fixed SPF after filling water into the ETS.

Let *d* be the diameter of the inserted rod, and let *D* be the diameter of the ETS. The SPF *η* is calculated as follows:(9)$$\eta =\frac{Nd^{2}}{D^{2}}\times 100\%$$
where *N* is the number of rods.

(2) *Indirect and direct data*. The data from the vacuum pressure meter on the pipe (see Figure 3a) can lead to an indirect label of the SPF for all the ETS measurements. These labels are abundant and available under all ETS working states, but often are inaccurate and erroneous. Alternatively, the direct data of the solid–liquid mixture in the pipe can be collected as a label, and then the corresponding SPF is measured through a balance, as shown in Figure 3b. Such sampling data are accurate, but their obtainable amounts are limited.

Figure 3c shows the comparison between the vacuum pressure and sampling data. As seen, the trend of the vacuum pressure data is roughly the same as that of the sampling data, but there is still a considerable number of errors between them. The sampling data are discontinuous, but they can be considered as accurate and standard labels. The vacuum pressure data are continuously collected by the meter, which may generate errors when directly using them for the calibration of the ETS.

To address this issue, we propose a data calibration method based on a mandatory-constraint FCM (MFCM) clustering algorithm, which is used to decrease the number of errors from indirect data, as explained below.

### 3.2. Cluster Characteristics of Sample Data

Let *D*_1_ be the set of *n* samples with erroneous and inaccurate labels as follows:(10)$$D_{1}=\{(X_{k},\eta_{k})|X_{k}\in R^{d},\eta_{k}\in R^{1},k=1,2,\ldots ,n\}$$
where ${\vec{X}}_{k}$ is the input vector with *d* variables (e.g., 240 measurements in the ETS), and *η_k_* is its corresponding label (e.g., the SPF).

Let *D*_2_ be the set of *Q* samples with accurate labels as follows:(11)$$D_{2}=\{(X_{q},\eta_{q})|X_{q}\in R^{d},\eta_{q}\in R^{1},\;q=1,2,\ldots ,Q\}$$
where ${\vec{X}}_{q}$ is the input vector with *d* variables, and *η_q_* is its corresponding accurate label (e.g., sampling data).

Since the label of the SPF mainly ranges in the interval of [0, 0.40], we partition the interval into six subintervals as follows: 0, [0.01, 0.1], [0.11, 0.20], [0.21, 0.30], [0.31, 0.40], and [0.41, 1.0]. Denote the set of input vectors on *D*_1_ and *D_2_* as follows:(12)$$S_{1}=\{X_{k}|X_{k}\in R^{d},\;k=1,\;2,\ldots ,n\}\;and\;S_{2}=\{X_{q}|X_{q}\in R^{d},q=1,\;2,\ldots ,Q\}$$

Let *S* = *S*_1_∪*S*_2_, and partition *S* into six clusters by the FCM algorithm. According to the CBR principle, the six clusters should correspond one-to-one to the six relative intervals of the labels, respectively, i.e., all the labels in each cluster must only fall into the interval. Since these data in *D*_1_ have erroneous and inaccurate labels, partial data must not be included in their relative intervals. To visually evaluate the consistency from the input to the output, we use the MDS (multidimensional scaling) [24] technique to map all the data in *S* to a two-dimensional space. MDS can preserve any between-point distances that are unchangeable from the high-dimensional data space to a selected low-dimensional data space. In particular, if the high dimension is not too large, the mapped distance is nearly unchangeable.

The data to be analyzed are a set of vectors *S* = {*X*_1_, *X*_2_, …, *X_n_*} in *R^d^* for which the distance function is defined as *d_ij_* = ||*X_i_*−*X_j_*|| for the *i*th and *j*th vectors. These distances consist of a dissimilarity matrix *D* = {*d_ij_*}∈*R^n^*^×*n*^. In view of *D*, the MDS aims to find a pair of vectors *Y_i_* and *Y_j_* in *R^2^* for any pair of vectors in *R^d^* such that the following is true:*d_ij_* = ||*X_i_* − *X_j_*|| = ||*Y_i_* − *Y_j_*|| for all *X_i_* and *X_j_*∈*S*
(13)
where || ● || is a vector norm. In a typical MDS, the norm is the Euclidean distance. Usually, the MDS is formulated as an optimization problem, where *Y*_1_, *Y*_2_, …, *Y_n_* are solved by the following typical cost function:(14)$$\min_{Y_{1},Y_{2},\ldots ,Y_{n}}{\left\{d_{ij}-||Y_{i}-Y_{j}||\right\}}^{2}$$

A solution may then be found by numerical optimization techniques. In this paper, the minimization solution is found in terms of the most used matrix eigenvalue decompositions [25].

After applying the MDS to *S*, each sample with the correct label (i.e., SPF *η*) in each cluster is marked as a red point, and the others are marked as blue circles. Table 1 shows the rates of samples that fall into their relative labeling intervals.

### 3.3. Mandatory Constraint Fuzzy Clustering for Calibration

To decrease the labeling errors in *D*_1_ by the accurate labels in *D*_2_, the objective function is defined as follows:(15)$$\min\;J_{2}=\sum_{i=1}^{c}\sum_{k=1}^{n}u_{ik}^{m}d_{ik}^{2}+\sum_{i=1}^{c}\sum_{q=1}^{Q}u_{iq}^{m}d_{iq}^{2}\;s.t.,\sum_{i=1}^{c}u_{ik}=1-\varepsilon ,\sum_{i=1}^{c}u_{iq}=\varepsilon$$
where $d_{ik}=||{\vec{X}}_{k}-{\vec{v}}_{i}|{|}_{2}$ and $d_{iq}=||{\vec{X}}_{q}-{\vec{v}}_{i}|{|}_{2}$; $u_{ik}$ and $u_{iq}$ are the membership degrees to *v_i_*; *i* = 1, 2, …, *c*; *k* = 1, 2, …, *n*; and *j* = 1, 2, …, *Q*. The value of *ε* represents the effect of these samples with accurate labels. Since the sum the membership degrees of an object for all clusters is 1, the sum of *Q* objects over all clusters in *D*_2_ has a maximum value *Q*. Hence, *ε*∈[0, *Q*], and 0 represents that the samples in *D*_2_ are not used.

The first term in Equation (15) is just the objective function of the FCM, while the second item stands for a mandatory constraint. Equation (15) specifies that any cluster center must not only minimize the sum of the distances to all points in *D*_1_ but also minimize the sum to all points in *D*_2_. *ε* is used to adjust the relative importance between the two items.

To minimize Equation (15), the Lagrange multiplier method [26] can transform it into the following equation:(16)$$L=\sum_{i=1}^{c}\sum_{k=1}^{n}u_{ik}^{m}d_{ik}^{2}+\sum_{i=1}^{c}\sum_{q=1}^{Q}u_{iq}^{m}d_{iq}^{2}+\sum_{k=1}^{n}\lambda_{k}(\sum_{i=1}^{c}u_{ik}-1+\varepsilon )+\sum_{q=1}^{Q}\mu_{q}(\sum_{q=1}^{c}u_{iq}^{\;}-\varepsilon )$$

The minimization of Equation (16) is usually based on the principle of alternating optimization, which involves solving the following two alternate problems.

Problem 1: Fix center *v_i_* to find the optimal membership degrees *u_ik_* and *u_iq_*, where *i* = 1, 2, …, *c* and *q* = 1, 2, …, *Q*.

Problem 2: Fix membership degrees *u_ik_* and *u_iq_* to find the optimal cluster center *v_i_*, where *i* = 1, 2, …, *c*.

For Problem 1, we take the partial derivative of the sum of the two ends in Equation (16) and let them be zero, as shown as follows:(17)$$\partial L/\partial u_{ik}=\sum_{k=1}^{n}mu_{ik}^{m-1}d_{ik}^{2}+\lambda_{k}=0$$
(18)$$\partial L/\partial u_{iq}=\sum_{q=1}^{Q}mu_{iq}^{m-1}d_{iq}^{2}+\lambda_{q}=0$$

From Equations (17) and (18), both *u_ik_* and *u_iq_* are solved as follows:(19)$$u_{ik}={\left(-\lambda_{k}/(md_{ik}^{2})\right)}^{1/(m-1)}and\;u_{iq}={\left(-\lambda_{q}/(md_{iq}^{2})\right)}^{1/(m-1)}$$

Since
(20)$$\sum_{t=1}^{c}u_{tk}=1-\varepsilon \;\;and\;\;\sum_{s=1}^{c}u_{sq}=\varepsilon$$

Thus, we insert Equation (19) into (20) and obtain the following:(21)$${\left(-\lambda_{k}/m\right)}^{1/(m-1)}=(1-\varepsilon )/\sum_{t=1}^{c}{\left(1/d_{it}^{2}\right)}^{{\;}^{1/(m-1)}}and\;{\left(-\lambda_{s}/m\right)}^{1/(m-1)}=\varepsilon /\sum_{s=1}^{c}{\left(1/d_{is}^{2}\right)}^{{\;}^{1/(m-1)}}$$

Insert Equation (21) back into (19) and obtain the following:(22)$$u_{ik}=(1-\varepsilon )/[\sum_{t=1}^{c}{\left(d_{ik}^{2}/d_{tk}^{2}\right)}^{1/(m-1)}],\;k=1,\;2,\ldots ,n$$
(23)$$u_{iq}=\varepsilon /[\sum_{s=1}^{c}{\left(d_{iq}^{2}/d_{sq}^{2}\right)}^{1/(m-1)}],\;\;q=1,\;2,\ldots ,Q$$

The process of solving Problem 2 is as follows. After taking the partial derivative of *v_i_* at both ends of Equation (16) and making it equal to zero, the following are derived:(24)$$v_{i}=\frac{\sum_{k=1}^{n}u_{ik}^{m}x_{k}+\sum_{q=1}^{Q}u_{iq}^{m}x_{q}}{\sum_{k=1}^{n}u_{ik}^{m}+\sum_{q=1}^{Q}u_{iq}^{m}},\;i=\;1,\;2,\ldots c$$

Let $v_{i}^{0}$ be the center when partitioning all data in *S*_1_ by FCM; $v_{i}^{0}$ must be different from $v_{i}$, and their difference is affected by the value of *ε.* When $\varepsilon =|D_{1}|/(|D_{1}|+|D_{2}|)$, it is a balancing point. Since the amount of data in *S*_2_ is very small, the difference between $v_{i}^{0}$ and $v_{i}$ is rather small, where *i =* 1, 2, …, *c*. To stress the effect of the data in *S*_2_, ε must be taken as larger than 0.5.

All samples in *D*_1_ are partitioned individually by FCM and MFCM, whereby two membership degrees $u_{ik}$ and $u_{ik}^{0}$ are obtained to *c* clustering centers, where *i =* 1, 2, …, *c*. Their differences are regarded as the weighting values to correct the label of the data in *D*_1_. Hence, the label of *X_j_* in *D*_1_ is corrected by the following coefficient:(25)$$h_{k}=\sum_{i=1}^{c}\omega_{i}(\frac{u_{ik}^{\;}}{u_{ik}^{0}}-1),\;k=1,\;2,\ldots ,n$$
where *ω_i_* is a normalized coefficient. And the label of any sample in *D*_1_ is corrected as
(26)$${\hat{\eta }}_{k}=\eta_{k}(1+\varphi h_{k}),\;k=1,2,\ldots ,n$$
where φ is a priori information on the value of *ε*. ${\hat{\eta }}_{k}$ is the new label of the *k*th sample in *D*_1_. The correcting process is shown in Figure 4.

By using the MFCM, the label of the vacuum pressure data in *D*_1_ is corrected. The comparison curves before and after the correction are shown in Figure 5.

Obviously, the trend of the corrected labels in *D*_1_ is closer to that of the sampling calibration data in *D*_2_ (see Figure 3c). After correcting all the labels in *D*_1_, the average absolute error of the corrected vacuum pressure data is decreased from 5.05% to 2.18%, and the average relative error is decreased from 17.44% to 6.23%.

Table 2 further shows the rate of correct labels in *D*_1_ before and after correction by the MFCM. The rate of data with the correct label at each cluster increased after the correction. The results further validate the effectiveness of the MFCM.

## 4. Experimental Section

### 4.1. Experimental Platform and Measuring Condition

The ETS measurements in the experiments come from data collected on February 2, 2023 at the Tianjin Bureau Dredging Experimental Platform, as shown in Figure 6a. The liquid in pipe is seawater with a conductivity of about 32 mS/cm, and the measured solid objects are fine sands. The set of indirect data with SPF labels from the vacuum pressure meter can be obtained, but the labels may have significant errors when estimating the SPF. Alternatively, since the experimental pipeline is horizontally closed in circulatory flow, and two-phase solid–liquid flow is evenly distributed in each cross-sectional pipe. Hence, SPF can be estimated by the rate between the added solid volume and the entire pipeline volume. Different rates of solid volumes will generate different SPFs, which are rather accurate and can be used for the accurate labelling of SPF. Therefore, the samples with accurate labels are used to decrease the error in the data from the vacuum pressure meter by MFCM.

The ETS can obtain 80 measurements a second under excitement frequency of 33.5 kHz and voltage of 10 Vpp. A total of 67,089 data from the vacuum pressure meter and the relative measurements from ETS were collected. After removing obvious anomalies and insufficient data, there were still 42,000 data. The SPF label of these data ranges from 0 to 29%. The entire interval was divided into 6 subintervals, as shown in Table 3.

Alternatively, 3000 data with various rates of solid-object volumes are obtained, which consist of a set of mandatory constraints with accurate labels. After calibrating ETS, the linear prediction model (LPM) based on Equation (6) is used to predict the SPF value. The following error criteria can be used to evaluate the predicting accuracy [27].

(1) *Root Mean Square Error*: the root mean squared error (RMSE) is a statistical indicator used to measure the deviation between the predicted value ${\hat{y}}_{i}$ and the true value *y_i_*; the closer the value is to 0, the more accurate the prediction is. For *N* samples, the calculation formula of RMSE is as follows:(27)$$RMSE=(\sum_{i=1}^{N}{\left({\hat{y}}_{i}-y_{i}\right)}^{2}/N{)}^{1/2}$$

(2) *Average absolute error*: the mean absolute error (MAE) is a very intuitive evaluation criterion that expresses the distance between the true and the predicted value. Like RMSE, MAE measures the absolute deviation between the true and the predicted value. Similarly, the closer it is to 0, the better the prediction effect. The MAE formula is as follows:(28)$$MAE=\sum_{i=1}^{N}|{\hat{y}}_{i}-y_{i}|/N$$

(3) *Average absolute percentage error*: The mean absolute percentage error (MAPE) normalizes the error of each point, making it less susceptible to extreme values and reducing its sensitivity to outlier data. The smaller the value, the better the prediction results. The calculation formula for MAPE is as follows:(29)$$MAE=\sum_{i=1}^{N}(|{\hat{y}}_{i}-y_{i}|/|y_{i}|)/N$$

(4) *Sample decision coefficient*: The coefficient of determination (*R*^2^) is a statistical indicator to reflect the reliability of the dependent variable. The purpose of the indicator is to test the explanatory power of any prediction model. The closer *R*^2^ is to 1, the closer the predicted value is to the true value. The calculation formula of *R*^2^ is as follows:(30)$$R^{2}=1-\sum_{i=1}^{N}{\left(y_{i}-{\hat{y}}_{i}\right)}^{2}/\sum_{i=1}^{N}{\left({\hat{y}}_{i}-{\bar{y}}_{i}\right)}^{2}$$

### 4.2. Experimental Results and Analysis

The experimental data are divided into two sets for ETS calibration by MFCM and for ETS prediction by LPM with a ratio of 0.7:0.3, where *λ* in the LPM algorithm is taken as 10^−5^, *m* = 1.5, and *ε* is taken as 0.60. Figure 7 shows the comparable curves of the prediction values by LPM after using correcting and noncorrecting labels by MFCM.

Figure 7 shows that after using the MFCM algorithm to correct the data labels, the LPM algorithm obtains more accurate SPFs and smaller errors, whereas the original maximum absolute error of the predicted values is about 10%. Moreover, a considerable portion of the relative error values reaches over 30% by noncorrected labels. After calibrating by corrected label, the absolute error of most of the predicted values is below 4 percentage points, with a maximum absolute error of about 8 percentage points and most of the relative error values below 30%.

Table 4 presents the four errors of RMSE, MAE, MAPE, and *R*^2^ when using the LPM for predictions with noncorrected and corrected labels by MFCM.

All four indexes show that the prediction accuracies of LPM have improved to some extent. The change in RMSE is more noteworthy, as this indicator is more sensitive to certain outliers, and its decrease indicates an improvement in the LPM algorithm to resist outliers. It is worth noting that both algorithms have high MAPE indicators, especially the linear regression model, which reaches 142.36% before calibration. This is mainly because the LPM is essentially a linear fitting of nonlinear data, with poor fitting degree and large absolute error at low SPF. But MAPE was greatly reduced to 62.65% after using the corrected labels by MFCM.

## 5. Conclusions

A calibration method is proposed for electrical tomography sensors based on fuzzy clustering with mandatory constraints. Using a small number of accurate labels as mandatory constraints, all inaccurate data are clustered and corrected to decrease the calibration error. By using the ratio of fuzzy membership degrees with and without mandatory constraints as the weighting value, the labels of all the inaccurate data are reclassified and calibrated. Our experimental results have shown that the new fuzzy clustering algorithm can effectively correct the labels of inaccurate data for ETS measurements. When the corrected data labels are used for predictions using the existing algorithm, the accuracy is greatly improved, providing a useful way to apply the ETS in practice. Furthermore, the proposed fuzzy clustering algorithm can be applied to the calibration process of any other sensor.

However, there are two issues that need to be solved in the future. One is how to determine the best objective function by selecting the value of ε, which can play an important role in the calibration process. The other involves the type of fuzzy clustering algorithm used. Any fuzzy clustering algorithm must be affected by its initiation and fuzzy exponents. How to find their optimal values remains a challenging task.

## Figures and Tables

**Figure 1 sensors-24-03068-f001:**
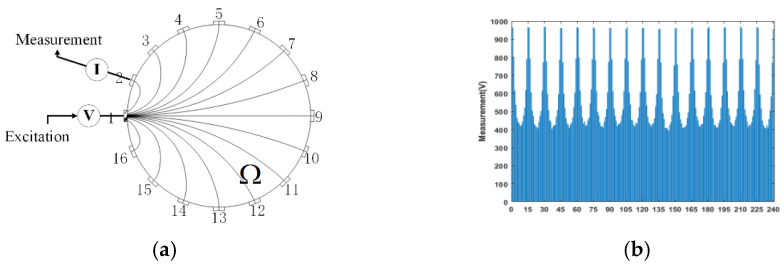
The ERT measuring process and all measurements from 16 electrodes. (**a**) Excitation and measurement of ERT; (**b**) 16 U-shape curves from 240 measurements.

**Figure 2 sensors-24-03068-f002:**
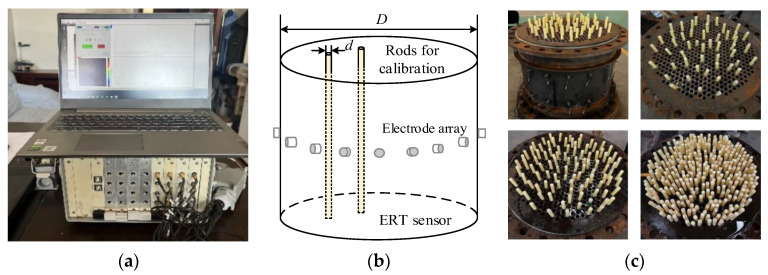
Ex-factory calibration of ETS. (**a**) Data acquisition device; (**b**) Calibration principle; (**c**) Different groups of rods.

**Figure 3 sensors-24-03068-f003:**
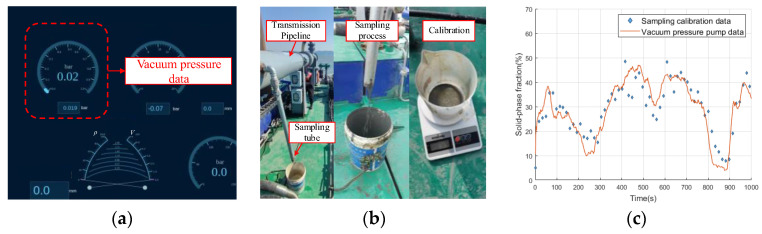
Indirect and direct calibration process. (**a**) Data from vacuum pressure meter; (**b**) Data from sampling; (**c**) Comparison of the two types of data.

**Figure 4 sensors-24-03068-f004:**
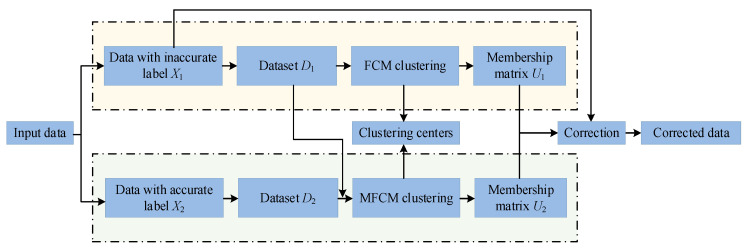
Flowchart for correcting the labels in *D*_1_ by MFCM.

**Figure 5 sensors-24-03068-f005:**
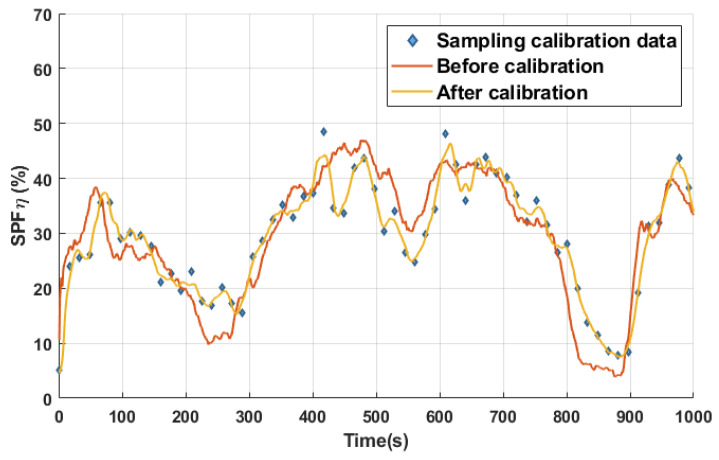
Comparison of error between corrected and non-corrected labeling data in *D*_1_.

**Figure 6 sensors-24-03068-f006:**
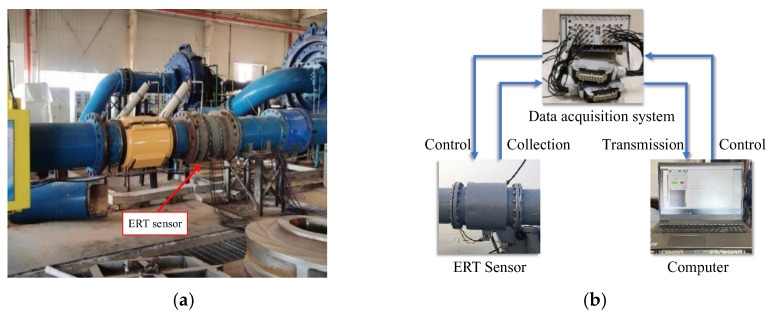
Experiment platform. (**a**) Sensors and pipeline in experiments; (**b**) Data acquisition system.

**Figure 7 sensors-24-03068-f007:**
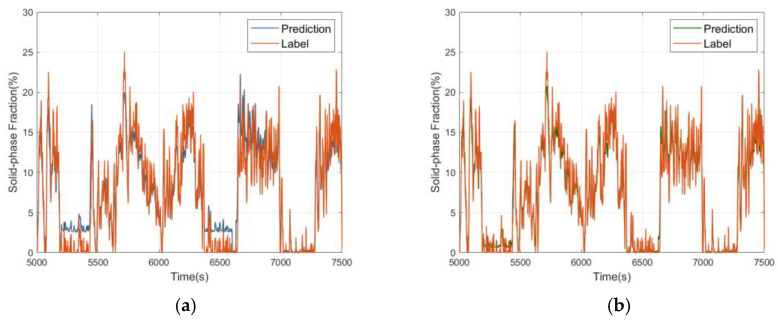
LPM for predicting SPF with corrected and non-corrected labels. (**a**) Prediction results using noncorrected labels; (**b**) Prediction results using corrected labels.

**Table 1 sensors-24-03068-t001:** Clustering results and the values of SPF *η* in six clusters and relative intervals.

0	[0.01, 0.10]	[0.11, 0.20]
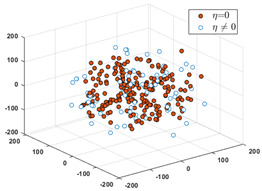 Dominant rate of SPF: 61.56%	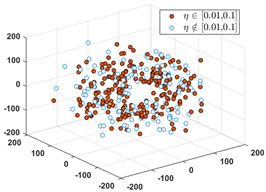 Dominant rate of SPF: 72.73%	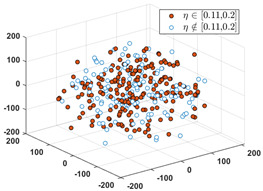 Dominant rate of SPF: 58.19%
[0.21, 0.30]	[0.31, 0.40]	[0.41, 1.00]
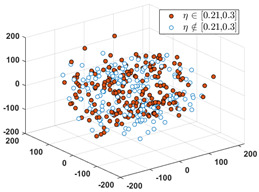 Dominant rate of SPF: 66.27%	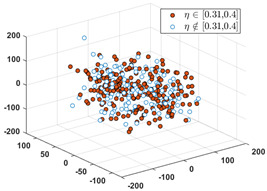 Dominant rate of SPF: 48.62%	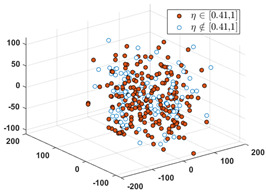 Dominant rate of SPF: 51.08%

**Table 2 sensors-24-03068-t002:** Comparing the number of correct labels between corrected and non-corrected data.

SPF Interval	0	[0, 0.10]	[0.11, 0.20]	[0.21, 0.30]	[0.31, 0.40]	[0.41, 1.00]
Noncorrected	61.56%	72.73%	58.19%	66.27%	48.62%	51.08%
Corrected	72.19%	79.49%	68.04%	72.16%	60.94%	58.35%

**Table 3 sensors-24-03068-t003:** Sample distribution of various SPFs.

SPF(%)	0~5	6~10	11~15	16~20	21~25	25~29	Total
Number	7000	7000	7000	7000	7000	7000	42,000

**Table 4 sensors-24-03068-t004:** Comparison of prediction errors by four indexes.

Index		RMSE	MAE	MAPE	R^2^
LPM	Noncorrected	2.6804	2.0141	142.36%	74.56%
	Corrected	1.8247	1.3137	62.65%	88.93%

## Data Availability

Data are contained within the article.

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
