# Peer review of "Electrical Sensor Calibration by Fuzzy Clustering with Mandatory Constraint"

_sensors, 2024, doi:10.3390/s24103068_

Round 1
Reviewer 1 Report
Comments and Suggestions for Authors
The overall impression is that this research is ambitious and worth posting. In this paper, the authors proposed a novel semi-supervised fuzzy clustering algorithm aimed at improving the accuracy of electrical tomography sensors for pipeline parameter detection and estimation. The algorithm leverages partial data with accurate labels to form mandatory constraints, thereby correcting errors in other labeling data. Through experiments conducted on a dredger, the authors validated the effectiveness of their proposed algorithm in terms of accuracy and stability. The significance of this work lies in its potential to address the inherent inaccuracies in sensor calibration caused by uncertain, inconsistent, incomplete, or invalid labeling data. Overall, the research presented in this paper demonstrates ambition and merits consideration for publication.
In general, I think the publication is well written, the subject interesting, up-to-date and presented clearly, and the scientific method appropriate. Thus, I recommend publication after the following points have been corrected/clarified.
1- The number of references provided in the manuscript is limited and should be expanded to provide a more comprehensive review of relevant literature. Increasing the number of references would strengthen the scholarly foundation of the paper and enhance its credibility within the academic community.
2- The introduction is too brief: While the introduction adequately presents the background information, it appears somewhat concise. Expanding the introduction would offer readers a more thorough understanding of the context and significance of the research. Additionally, ensuring that all pertinent references are included in the introduction would further enrich the scholarly basis of the paper.
3- The methods are adequately described. Several captions and figures require repositioning and adjustment, specifically to be relocated and aligned to the centre. Proper positioning of these elements is crucial for maintaining the visual coherence and organization of the manuscript. Therefore, it is recommended to make the necessary revisions to ensure that captions and figures are appropriately centered and situated within the middle pages, thereby enhancing the overall presentation of the paper.
4- Table 2, Figure 5: should be provided in vector graphics to be scalable! Otherwise, some numbers are difficult to read even after zooming in on the screen.
Author Response
Thankful for the reviewer's careful reading, and please see the reply in the attached.

Reviewer 2 Report
Comments and Suggestions for Authors
In the article, the authors present the problem of calibration of sensors in EIT. To decrease the calibration errors for erroneous and incorrect data the fuzzy clustering algorithm with constraints has been applied. The article is prepared using commonly used standards. The study requires explanations, for example:
- 132-134, what part of the total variation of training set is explained by two latent variables used to determine the vector $Y \in R^2$?
- 69-73, In EIT, the predictors in the learning set are usually correlated. The authors only use Tikhonov regularisation to reduce dimensionality. In EIT, we additionally use other techniques, such as LASSO, Elasticnet, to eliminate redundant predictors or extract latent variables from the dataset by applying PCA, e.g. Rymarczyk, T.; KÅ‚osowski, G.; KozÅ‚owski, E.; Tchórzewski, P. Comparison of Selected Machine Learning Algorithms for Industrial Electrical Tomography. Sensors 2019, 19, 1521. https://doi.org/10.3390/s19071521
Przysucha, B.; Wójcik, D.; Rymarczyk, T.; Król, K.; KozÅ‚owski, E.; GÄ…sior, M. Analysis of Reconstruction Energy Efficiency in EIT and ECT 3D Tomography Based on Elastic Net. Energies 2023, 16, 1490. https://doi.org/10.3390/en16031490
- Task (15) is an extension of the classical FCM algorithm. What does the value $\varepsilon$ in formula (15) mean? Give the range of possible values $\varepsilon$.
Author Response
Thanks for the reviewer's careful reading, and please the reply in the attached

Reviewer 3 Report
Comments and Suggestions for Authors
This paper presents the calibration of electrical sensors via fuzzy clustering with mandatory constraint.
The article is an interesting approach in the aspect of electrical tomography. Nevertheless, the purpose of this research in connection with the quality of image reconstruction when solving the inverse problem is unclear.
Minor remarks:
1) I propose to describe in more detail on what basis and what results from the scope of data selection for the research problem.
2) What features of the presented model are an original approach in the studied discipline
3) The authors could provide more data related to measurement parameters, current/voltage values used, frequency, and number of measurements per second.
4) What is the influence of the tested object on the calibration parameters?
5) The purpose and effectiveness of the calibration performed in terms of its impact on the quality of the reconstructed images are not clearly described. What do these studies show?
Author Response
Thanks for the reviewer's careful reading, and please see the reply in the attached.
